# A Yes-Associated Protein (YAP) and Insulin-Like Growth Factor 1 Receptor (IGF-1R) Signaling Loop Is Involved in Sorafenib Resistance in Hepatocellular Carcinoma

**DOI:** 10.3390/cancers13153812

**Published:** 2021-07-29

**Authors:** Mai-Huong T. Ngo, Sue-Wei Peng, Yung-Che Kuo, Chun-Yen Lin, Ming-Heng Wu, Chia-Hsien Chuang, Cheng-Xiang Kao, Han-Yin Jeng, Gee-Way Lin, Thai-Yen Ling, Te-Sheng Chang, Yen-Hua Huang

**Affiliations:** 1International Ph.D. Program for Cell Therapy and Regeneration Medicine, College of Medicine, Taipei Medical University, Taipei 11031, Taiwan; d151106002@tmu.edu.tw (M.-H.T.N.); d151109009@tmu.edu.tw (C.-X.K.); 2Department of Biochemistry and Molecular Cell Biology, School of Medicine, College of Medicine, Taipei Medical University, Taipei 11031, Taiwan; d119103012@tmu.edu.tw (S.-W.P.); gwl@tmu.edu.tw (G.-W.L.); 3TMU Research Center of Cell Therapy and Regeneration Medicine, Taipei Medical University, Taipei 11031, Taiwan; s03271@tmu.edu.tw (Y.-C.K.); m609108001@tmu.edu.tw (H.-Y.J.); 4Institute of Information Science, Academia Sinica, Taipei 11529, Taiwan; cylin@iis.sinica.edu.tw (C.-Y.L.); t050508@iis.sinics.edu.tw (C.-H.C.); 5The Ph.D. Program for Translational Medicine, College of Medical Science and Technology, Taipei Medical University, Taipei 11031, Taiwan; mhwu1015@tmu.edu.tw; 6Graduate Institute of Biomedical Informatics, College of Medical Science and Technology, Taipei Medical University, Taipei 11031, Taiwan; 7International Ph.D. Program for Translational Science, College of Medical Science and Technology, Taipei Medical University, Taipei 11031, Taiwan; 8Department of Pathology, Keck School of Medicine, University of Southern California, Los Angeles, CA 90033, USA; 9Department and Graduate Institute of Pharmacology, National Taiwan University, Taipei 100, Taiwan; tyling@ntu.edu.tw; 10School of Traditional Chinese Medicine, College of Medicine, Chang Gung University, Taoyuan 33382, Taiwan; 11Division of Gastroenterology and Hepatology, Department of Internal Medicine, Chang Gung Memorial Hospital, Chiayi 61363, Taiwan; 12Graduate Institute of Medical Sciences, College of Medicine, Taipei Medical University, Taipei 11031, Taiwan; 13Center for Reproductive Medicine, Taipei Medical University Hospital, Taipei 11031, Taiwan; 14Comprehensive Cancer Center of Taipei Medical University, Taipei 11031, Taiwan; 15TMU Research Center of Cancer Translational Medicine, Taipei Medical University, Taipei 11031, Taiwan

**Keywords:** YAP, IGF-1R, sorafenib resistance, EMT markers, HCC

## Abstract

**Simple Summary:**

Sorafenib is the first approved targeted therapy for the treatment of advanced hepatocellular carcinoma (HCC). However, HCC resistance to sorafenib has greatly reduced its utility. Yes-associated protein (YAP) is overexpressed in cancers, including HCC. We observed a positive correlation in expression levels of insulin-like growth factor-1 receptor (IGF-1R) and YAP in sorafenib-resistant HCCs. Therefore, the interplay between YAP and IGF-1R signaling and its role in HCC sorafenib resistance will be examined in this study. We found that the YAP-IGF-1R signaling loop was involved in sorafenib resistance in HCC. IGF-1/2 treatment enhanced YAP nuclear translocation. In turn, YAP regulated expression of IGF-1R and epithelial mesenchymal transition (EMT)-related proteins in vitro. Targeting YAP with a specific inhibitor, verteporfin (VP), significantly increased HCC cell sensitivity to sorafenib, with a potential synergistic combination index. These findings highlight the significance of the YAP-IGF-1R signaling loop as a potential therapeutic target for HCC, especially in terms of overcoming sorafenib resistance.

**Abstract:**

The role of a YAP-IGF-1R signaling loop in HCC resistance to sorafenib remains unknown. Method: Sorafenib-resistant cells were generated by treating naïve cells (HepG2215 and Hep3B) with sorafenib. Different cancer cell lines from databases were analyzed through the ONCOMINE web server. BIOSTORM–LIHC patient tissues (46 nonresponders and 21 responders to sorafenib) were used to compare YAP mRNA levels. The HepG2215_R-derived xenograft in SCID mice was used as an in vivo model. HCC tissues from a patient with sorafenib failure were used to examine differences in YAP and IGF-R signaling. Results: Positive associations exist among the levels of YAP, IGF-1R, and EMT markers in HCC tissues and the levels of these proteins increased with sorafenib failure, with a trend of tumor-margin distribution in vivo. Blocking YAP downregulated IGF-1R signaling-related proteins, while IGF-1/2 treatment enhanced the nuclear translocation of YAP in HCC cells through PI3K-mTOR regulation. The combination of YAP-specific inhibitor verteporfin (VP) and sorafenib effectively decreased cell viability in a synergistic manner, evidenced by the combination index (CI). Conclusion: A YAP-IGF-1R signaling loop may play a role in HCC sorafenib resistance and could provide novel potential targets for combination therapy with sorafenib to overcome drug resistance in HCC.

## 1. Introduction

Globally, the number of new liver cancer cases ranked seventh among all cancers, while deaths attributable to liver cancer ranked second, accounting for 8.2% of all cancer deaths in 2018 [1]. The gap between the incidence and mortality rates of liver cancer is due to the aggressiveness of the disease and the lack of effective treatments. Hepatocellular carcinoma (HCC) accounts for more than 90% of liver cancers. Drug resistance is the major reason for the unsatisfactory response to targeted therapies, including sorafenib and lenvatinib, in the treatment of advanced HCC. In the Phase 3 SHARP clinical trial and the REFLECT clinical trial, disease progression in the sorafenib treatment group was 27% and 31%, respectively [2,3]. The low response rate to sorafenib in HCC treatment is a current clinical challenge.

A number of studies have demonstrated various mechanisms by which cancer cells are activated in order to escape the lethality of sorafenib [4,5]. Factors found to be involved in sorafenib resistance include the signaling pathways PI3K/AKT [6,7], Raf/Mek/ERK [8], Jak/Stat3 [9,10], cancer stemness [11,12], hypoxic environments [13], epithelial mesenchymal transition (EMT) [14], microenvironmental and metabolic derangement [15], and autophagy [16]. Despite all these findings, the mechanism of sorafenib resistance in HCC is still not fully understood.

Yes-associated protein (YAP) is an effector of the Hippo pathway, which is well known as a tissue development and growth-regulating signaling pathway in both proliferation and apoptosis. YAP is also known to be an oncoprotein that promotes transcription for cell proliferation, cell survival, and anti-apoptosis. The dysregulation of YAP/transcriptional activator with PDZ domain (TAZ) drives tumor development, metastasis, and therapy resistance [17,18,19]. YAP/TAZ was shown to play a critical role in resisting a variety of targeted drugs [18], including erlotinib [20] and epidermal growth factor receptor (EGFR) tyrosine kinase inhibitors (erlotinib, gefitinib, afatinib, and AZD8931) in lung cancer [21], and B-RAF inhibitor (vemurafenib) and MEK inhibitor (trametinib) in human melanoma, colon, and thyroid cancer cell lines [22]. In liver cancer, YAP and TAZ were demonstrated to promote resistance to multiple drugs [23] and to mediate regorafenib efficacy [24]. Specifically, high expression of YAP was found in stiffness-induced sorafenib resistance [25] and in hypoxia-mediated sorafenib resistance [26] in HCC.

High expression of insulin-like growth factor 1 receptor (IGF-1R) is highly associated with cancer stemness [12,27,28] and sorafenib resistance [11] in HCC. YAP has been found to promote radioresistance and genomic instability in medulloblastoma through IGF2-mediated Akt activation [29]. Furthermore, a positive correlation between IGF1R and YAP was identified in hypoxic conditions [30]. In the current study, we demonstrate a critical role for YAP in HCC sorafenib resistance. Suppression of YAP downregulated expression of IGF-1R and EMT markers and conferred sorafenib resistant properties to HCC cell lines. Activation of IGF-1R by its ligands (IGFs) resulted in increased nuclear translocation of YAP. Nuclear translocation of YAP was inhibited when the IGF-1R phosphorylation inhibitor linsitinib was present. The findings of this study demonstrate a potential correlation between IGF1R and YAP in the development of sorafenib resistance and suggest that the combination of the YAP-specific inhibitor verteporfin and sorafenib may be an effective treatment for HCC.

## 2. Materials and Methods

### 2.1. Cell Lines and HCC Tissues

HepG2215 cells (HBV^+^/HBsAg^+^ human hepatoblastoma) were a gift from Dr. Jun-Jen Liu (Institute of Medical Biotechnology, Taipei Medical University, Taipei, Taiwan). Hep3B cells (HBV^+^ HBsAg^+^ human HCC, HB-8064) were purchased from the American Type Culture Collection (ATCC, Manassas, VA, USA). Sorafenib-resistant cells (HepG2215_R and Hep3B_R) were generated by treating naïve cells (HepG2215 and Hep3B) with sorafenib (Cell Signaling, Danvers, MA, USA) starting at a low concentration and proceeding to a high concentration. The sorafenib-resistant cells were then maintained in 10 µM sorafenib (Cell Signaling, Danvers, MA, USA). Dulbecco’s modified Eagle medium (DMEM, Gibco-BRL, Thermo Fisher Scientific, Waltham, MA, USA) plus 10% fetal bovine serum (FBS), 3.7 g/L sodium bicarbonate (Sigma-Aldrich, St. Louis, MO, USA), 1% penicillin-streptomycin (PS, Gibco, Grand Island, NY, USA), and 1% glutamate (Gibco, Grand Island, NY, USA) was used for culturing all cell lines.

HCC tissues were obtained from a patient who had received sorafenib after resection of a large HCC and had a peritoneal lymph node excised 2 months after sorafenib treatment. This study was approved by the Institutional Review Board of Chang Gung Medical Foundation (Approval number: 201800008B0C601).

### 2.2. BIOSTORM Patient Cohort and TCGA-LIHC Cohort

The microarray data of tumor tissue and clinical information from 67 patients with HCC, who received sorafenib, were download from the GSE109211 dataset in the BIOSTORM study [31]. Among them, 46 HCC patients did not respond and 21 HCC patients responded to sorafenib. The definition of patients responding to sorafenib was based on whether patients benefited from sorafenib in terms of extended recurrence-free survival.

The htseq-count files of TCGA-LIHC project [32] were downloaded from The National Cancer Institute (NCI) Genomics Data Commons (GDC) Data portal (https://portal.gdc.cancer.gov/ (accessed on 4 January 2021)). Based on sample clinical metadata, we selected the htseq-count samples from patients who were treated with sorafenib and only had hepatitis B factors in patient’s history hepatocarcinoma risk factors and viral hepatitis serology. Some samples met with our criterial; therefore, htseq-count samples from five primary tumor tissue and three normal adjacent tissue were normalized by TMM via edgeR package [33] in R environment.

### 2.3. Tumor Xenograft Mouse Model

The eight-week-old NOD-SCID mice (National Laboratory Animal Center, Taipei, Taiwan) were subcutaneously inoculated with HepG2215-R cells. Mice were injected with 5 × 10^6^ cells on the left flank. Sorafenib was given to the mice twice a week. After 8 weeks of implantation, the mice were sacrificed. Tumor tissues were collected and embedded in paraffin wax. Immunohistochemistry staining assays against YAP, IGF-1R, VIMENTIN, SNAIL1, N-CADHERIN, and E-CADHERIN were applied and analyzed. The animal study protocol was approved by the Institutional Animal Care and Use Committee/Panel at Taipei Medical University, Taipei, Taiwan (Approval number: LAC-2017-0433).

### 2.4. mRNA Extraction, cDNA Conversion, and Quantitative PCR

Total mRNA was extracted from cells and purified using the EasyPure Total RNA Spin Kit (Bioman Scientific, New Taipei, Taiwan) according to the manufacturer’s directions. The concentration of total mRNA was measured with the NanoPhotometer N60 system. cDNA was synthesized from mRNA using MMLV reverse transcriptase (Invitrogen, Carlsbad, CA, USA). Real-time PCR was performed using Fast SYBR™ Green Master Mix (Thermo Fisher Scientific, Waltham, MA, USA) to evaluate mRNA levels in the samples. Amplification was done with the StepOnePlus Real-Time PCR system (Applied Biosystems, Vilnius, Lithuania) and steps included denaturation: 95 °C for 30 s; and thermal cycles repeated 35 times: 95 °C for 3 s and 60 °C for 30 s. The GAPDH gene was used as an internal control. The primer sequences are listed in Appendix A.

### 2.5. Cell Growth Assay

Cells were seeded in 96-well plates (DB Falcon, Durham, NC, USA) at 5000 cells/well. Cell viability was measured by WST-1 reagent (Roche, Basel, Switzerland). WST-1 10% was added into cell culture wells for 1.5 h. The absorbance at 400 nm and 600 nm was measured by the SPARK 10M system (TECAN, Zurich, Switzerland).

### 2.6. Cytoplasmic/Nuclear Protein Extraction

Cell membranes were lysed in buffer A containing 10 mM KCl, 2 mM MgCl_2_, 1 mM DTT, 0.1 mM EDTA, 10 mM HEPES, pH 7.8, protease inhibitor cocktail (Roche), and phosphatase inhibitor cocktail (Roche) for 10 min on ice. The lysis solution was centrifuged at 4 °C and 14,000 rpm for 15 min. The supernatant (cytoplasmic components) was moved to a new Eppendorf 1.5 mL tube and the pellet (nuclear components) was washed two times with buffer A before the nuclear envelope was lysed using ultrasound in buffer C, which contained 1 mM KCl, 1 mM DTT, 0.1 mM EDTA, 50 mM HEPES, pH 7.8, 300 mM NaCl, 20% glycerol, protease inhibitor cocktail (Roche), and phosphatase inhibitor cocktail (Roche). The nuclear proteins were collected by centrifuging at 4 °C and 14,000 rpm for 15 min.

### 2.7. Western Blotting

Cells were lysed by RIPA buffer (Energenesis Biomedical, Taipei, Taiwan) with protease inhibitor cocktail (Roche) and phosphatase inhibitor cocktail (Roche). The proteins were collected using centrifugation at 4 °C and 14,000 rpm for 25 min. The protein concentration was assessed by the Pierce™ BCA assay kit (Thermo Fisher Scientific, Waltham, MA, USA) according to the manufacturer’s instructions. Equal amounts of protein samples were separated by 10–12% SDS-PAGE and transferred onto nitrocellulose membranes (Immobilon, Merck Millipore, Darmstadt, Gemany). Specific antibodies were used to indicate the presence of target proteins (concentrations and information for each antibody is shown in Appendix A). The signal of the complex of target protein-primary antibody-secondary antibody was detected using the ImageQuant LAS 4000 mini system (GE Healthcare, Chicago, IL, USA). For all Western blot figures, we included densitometry readings/intensity ratio of each band in Appendix A. In addition, we also included some whole blots showing all the bands with all molecular weight markers (Appendix A).

### 2.8. Immunocytochemistry Fluorescence Staining

Cells were seeded on coverslips in six-well plates. After culturing and drug treatments, cells were fixed with 4% paraformaldehyde for 20 min. Permeabilization was done with PBS plus 0.2% Tween-20 for 5 min. Primary and secondary antibodies were prepared in PBS plus 1% FBS. The secondary antibody was Alexa-488 (Thermo Fisher Scientific, Waltham, MA, USA), which showed a fluorescent green. Nuclei were stained with DAPI (1 μg/mL) for 5 min. PBS was used for every washing step.

### 2.9. Immunohistochemistry

Tumor slides were incubated at 65 °C for 1 h before deparaffinization by xylene followed by washing with decreasing concentrations of ethanol (from 100% to 60%) and then water. Slides were then soaked in boiled 1% unmasking solution for 30 min. After cooling, the slides were soaked in 3% H_2_O_2_ in PBS at room temperature for 15 min. Blocking was done using Normal Goat serum, 2.5% (ImmPRESS^®^ Goat Anti-Rabbit IgG polymer Kit–Vector [MP-7451]) for 1 h before incubation with primary antibodies at 4 °C overnight. Slides were incubated with secondary antibody (ImmPRESS^®^ Goat Anti-Rabbit IgG polymer Kit–Vector [MP-7451]) for 1 h at room temperature. The DAB Substrate Kit–Vector (SK4100) was used. Nuclei were stained with Hematoxylin. Finally, slides were dehydrated using increasing concentrations of ethanol and xylene before being mounted using Malinol reagent.

### 2.10. Short Hairpin RNA and Plasmids

The packaging pCMVΔR8.91 plasmid and the envelope VSV-G pMD.G plasmid were co-transfected with shYAP#1 (TRCN0000107265), shYAP#2 (TRCN0000300281), shYAP#3 (TRCN0000300325), shIGF-1R#1 (TRCN0000000425), shIGF-1R#2 (TRCN0000000426), or shLacZ (TRCN0000072260) plasmids (Taiwan RNAi consortium, Taipei, Taiwan) into HEK293T cells using Turbofect transfection reagent according to the manufacturer’s instructions (Thermo Fisher Scientific, Waltham, MA, USA).

### 2.11. The Drug Combination Index (CI)

Cells were seeded into 96-well plates (BD Falcon, Durham, NC, USA) at 5000 cells/well. Cells were treated with a single drug or a combination. Concentrations of sorafenib were 0, 1.25, 2.5, 5, 7.5, 10, and 15 μM. Concentrations of verteporfin were 0, 0.5, 1, and 1.5 μg/mL. Cell viability was measured using the WST1 assay. The drug combination index was analyzed by Compusyn software. CI values of <0.9, 0.9–1.1, and >1.1 were considered as synergism, additive effects, and antagonism, respectively.

### 2.12. Statistical Analysis

Data are presented as standard error of mean (SEM). Statistical differences in mean were analyzed by Student’s *t*-test or Mann–Whitney U-test. Pearson’s χ^2^ test was used to examine the relationship among different gene and protein levels (GraphPad Software, La Jolla, CA, USA). A *p*-value < 0.05 was considered statistically significant.

## 3. Results

### 3.1. High Expression Levels of YAP Correlate with Sorafenib-Resistant Properties of HCCs

A critical role for YAP in cancer drug resistance has recently been addressed [18]; however, the role of YAP in sorafenib resistance in HCC is still unclear. To study the clinical association between YAP and HCC sorafenib resistance, we used the ONCOMINE webserver to examine mRNA levels of *YAP* in both cancer cells and normal cells of different cancers. As shown in Figure 1, compared with levels in normal cells, significantly higher expression levels of *YAP* were observed in most of the solid tumors, including the liver cancer, in both the Barretina CellLine (left panel) and Garnett CellLine (right panel) databases (Figure 1A). Furthermore, *YAP* expression levels were significantly higher in tumors from HCC patients who did not respond to sorafenib (non-responders, *n* = 46) than that of in HCC patients who responded to sorafenib (responders, *n* = 21, BIOSTORM–LIHC patient tissues) (Figure 1B).

The role of YAP in HCC sorafenib resistance was further examined using sorafenib-resistant HepG2215 and Hep3B cells (HepG2215_R and Hep3B_R) [11]. When compared with naïve cells, both sorafenib-resistant cell lines had higher IC50 values (Figure 1C). In addition, levels of YAP protein in sorafenib-resistant HCCs were significantly higher than those in sorafenib-naïve cells (Figure 1D). These results strongly suggest a potential role for YAP in sorafenib resistance in HCC cells.

To further examine the role of YAP in the sorafenib resistance of HCC cells, a YAP-specific inhibitor, verteporfin (VP), was used to suppress YAP transcriptional activity and the IC50 values of sorafenib-resistant cells with or without VP (0.5, 1, and 1.5 μg/mL) were determined. We found that VP significantly increased sorafenib sensitivity in a dose-dependent manner in both the HepG2215_R and Hep3B_R cells (Figure 1E). Meanwhile, the combination of VP and sorafenib showed a synergistic effect, suppressing cell viability of both sorafenib-resistant HepG2215_R and Hep3B_R cells (Figure 1F). The CI values of VP and sorafenib are shown in Appendix A. Together, these results demonstrate an important role for YAP in sorafenib-resistant HCC.

### 3.2. YAP Is Highly Associated with IGF-1R and EMT-Related Proteins in Sorafenib-Resistant HCCs

We have previously demonstrated that IGF-1R activation promotes the expression of stemness-related properties in HBV-related HCCs, and confers poor prognosis in patients [27]. Interestingly, YAP has been shown to promote radioresistance and genomic instability in medulloblastoma through IGF-2-mediated AKT activation [29]. We thus examined if a correlation exists between YAP and IGF-1R/EMT-related proteins in sorafenib-resistant HCC cells. For these experiments, sorafenib-resistant HepG2215_R cells were used in both the in vitro cell model and in vivo animal model. As shown in Figure 2, compared with naïve cells, sorafenib-resistant HCC cells had significantly higher expression levels of YAP, as well as of IGF-1R; IGF-1; and the mesenchymal-related proteins VIMENTIN, SNAIL1, and N-CAD, both at the transcriptional level, as shown by mRNA analysis (Figure 2A and Appendix A), and at the protein level, as shown by Western blotting analysis (Figure 2B).

These data were further verified by an in vivo animal study and immunohistochemical staining. As shown in Figure 2C, YAP, IGF-1R, and EMT markers (VIMENTIN, SNAIL1, and N-CAD) were visibly expressed in tumor tissues. Notably, there was a high percentage of cells with nuclear YAP expression. Meanwhile, YAP, IGF-1R, and mesenchymal markers (VIMENTIN and SNAIL1) were consistently observed at the tumor margin. In contrast, the expression level of E-CAD was low in sorafenib-resistant HCCs. These data strongly support a high correlation between YAP and IGF-1R and the mesenchymal proteins (VIMENTIN, SNAIL1, and N-CAD) in sorafenib-resistant HCCs.

### 3.3. YAP Regulates IGF-1R Signaling-Related Proteins and EMT Markers in Sorafenib-Resistant HCCs

YAP has been demonstrated to regulate the expression of EMT-related proteins in various cancers including non-small cell lung cancer [34,35], intrahepatic cholangiocarcinoma [36], colorectal cancer [37,38], renal cancer [39], laryngeal cancer [40], and esophageal squamous cell cancer [41]. To examine the role of YAP in the expression of IGF-1R signaling and EMT markers in sorafenib-resistant HCCs, we used the YAP-specific inhibitor VP and RNA silencing shYAP plasmids. As shown in Figure 3, VP significantly suppressed mRNA levels of *YAP*, *IGF-1R*, *VIMENTIN*, *SNAIL1*, and *N-CAD* in a dose-dependent manner in both HepG2215_R and Hep3B_R cells (Figure 3A). Western blotting results further demonstrated the suppressive effect of VP on the protein levels of YAP, IGF-1R, and its downstream signal proteins AKT and the EMT markers VIMENTIN, SNAIL1, and N-CAD (Figure 3B). The quantitative results of Figure 3B are shown in Figure 3C. These observations were further supported using an shYAP RNA silencing strategy (Figure 3D). VP treatment also suppressed the expression of IGF-1R in sorafenib naïve cells (HepG2215 and Hep3B cells) (Appendix A).

### 3.4. IGF-1R Activation Induces YAP Nuclear Translocation in Sorafenib-Resistant HCC Cells

It has been well documented that the activated form of YAP (non-phosphorylated form) translocates into the nucleus, binds to nuclear co-transcription factors, and initiates gene transcription [42,43,44,45]. Thus, the presence of nuclear YAP is highly associated with cellular activities. To examine the nuclear localization of YAP protein in sorafenib-naïve and sorafenib-resistant HBV-HCCs (HepG2215 and HepG2215_R), immunocytochemical staining was performed. As shown in Figure 4, compared with naïve HepG2215 cells, sorafenib-resistant HepG2215_R cells had a significantly higher percentage of cells with nuclear YAP staining. The percentage of cells with nuclear YAP was 40.8% and 1.8% for HepG2215_R cells and HepG2215 cells, respectively (Figure 4A). These results demonstrate the dominant nuclear YAP population in sorafenib-resistant HepG2215_R cells.

It is also well documented that activation of IGF-1R signaling is associated with a high recurrence rate of HBV-related HCC [27]. Therefore, we examined the effects of IGF1/2 on YAP nuclear translocation in sorafenib-resistant HepG2215_R cells. As shown in Figure 4, IGF-1/2 treatment (1 h or 2 h) effectively induced YAP nuclear translocation in these cells (Figure 4B). The levels of YAP protein in the nuclei were measured and sorted into three categories according to fluorescence intensity: non group (fluorescence intensity value < 500), low group (fluorescence intensity value 500–1200), and high group (fluorescence intensity value > 1200) (Figure 4C). Actively dividing cells were excluded from categorization. Compared with the control group, the IGF-1/2 groups showed a significantly higher percentage of cells with high nuclear-YAP expression (Figure 4D). The effect of IGF-1/2 on YAP nuclear translocation in naïve HepG2215 is shown in Appendix A.

The role of IGF-1R signaling in the nuclear translocation of YAP was further demonstrated by Western blotting, which showed that IGF-1/2 treatment effectively increased the total YAP protein level in sorafenib-resistant HepG2215_R cells (Figure 4E). Furthermore, silencing IGF-1R by shIGF-1R significantly decreased YAP protein levels (Figure 4F,G). These observations were further verified using linsitinib, a specific inhibitor of IGF-1R activation. We found that linsitinib treatment significantly suppressed IGF-1/2-induced YAP protein levels in the nuclei of sorafenib-resistant HepG2215_R cells (Figure 4H).

To examine the role of the IGF-1R downstream signaling proteins PI3K/AKT–mTOR–ERK in YAP nuclear translocation, specific inhibitors targeting PI3K (LY294002), mTOR (rapamycin), and ERK (PD98059) were used in combination with IGF-1/2 treatment in sorafenib-resistant HepG2215_R cells. As shown in Figure 4I, LY294002 and rapamycin effectively suppressed IGF-1-induced YAP nuclear translocation in HepG2215_R cells, but no effect with PD98059 was observed (Figure 4I). Together, these results demonstrate a role for IGF-1R–PI3K/AKT–mTOR signaling in YAP expression and nuclear translocation in sorafenib-resistant HCCs.

### 3.5. YAP Expression Is Positively Correlated with IGF-1R and EMT-Related Proteins in Tumor Tissues from HCC Patients

To examine the correlation of YAP with IGF-1R and EMT markers in clinical HCC tissue, the TCGA-HCC cohort was analyzed using the GEPIA webserver. As shown in Figure 5A, mRNA levels of *YAP* were significantly and positively correlated with those of *IGF-1R* (*R* = 0.47, *p* < 0.0001), *VIMENTIN* (*R* = 0.24, *p* = 2.8 × 10^−6^), *SNAIL1* (*R* = 0.2, *p* = 0.00012), and *E-CAD* (also known as *CDH1*, *R*= 0.32, *p* = 3.1 × 10^−10^). To further investigate the association of YAP with IGF-1R, VIMENTIN, and SNAIL1 in human HCC, we used treatment-naïve HCC tissue from a patient who received resection of a ruptured HCC at Chang-Gung Memorial Hospital, Chiayi, Taiwan. Sorafenib treatment was initiated after HCC resection. Two months later, the patient received another surgery to remove a new peritoneal metastatic lymph node. We considered this lymph node as the sorafenib-resistant HCC. Immunohistochemical staining of the HCC tissue (pre-treatment) and of the metastatic tissue (post-treatment) was performed. As shown in Figure 5B, we found high expression levels of nuclear YAP protein as well as IGF-1R and EMT markers in the HCC tissue (Figure 5C). Notably, high expression levels of nuclear YAP protein, IGF-1R, VIMENTIN, and SNAIL1 were detected in the post-treatment metastatic tissue (Figure 5C). Furthermore, the strong and positive correlations among mRNA expression levels of *YAP*, *IGF-1R* (*R*^2^ = 0.8530, *p* < 0.0011), and *IGF-2* (a liver pathology factor, *R*^2^ = 0.7585, *p* < 0.0049) in the sorafenib treatment TCGA cohort were observed (Figure 5D,E). These data strongly support a positive correlation among nuclear YAP expression, IGF-1R, and EMT markers in HCC patients, and suggest a potential role for a YAP–IGF-1R signaling loop in the sorafenib resistance of HCC.

## 4. Discussion

Approved in 2007, sorafenib was the only systemic agent with proven clinical efficacy for patients with unresectable HCC until the recent development of several new targeted therapies and immunotherapies. The high frequency of sorafenib resistance has significantly limited its utility in the treatment of HCC. Based on the public database demonstrating a high level of *YAP* expression in many solid cancer cell lines (Figure 1A), we examined *YAP* mRNA levels in sorafenib nonresponders and found that *YAP* was more elevated than in sorafenib responders (Figure 1B). In addition, inhibiting YAP with a specific inhibitor (verteporfin) sensitized the sorafenib-resistant cell lines to sorafenib (Figure 1E). None of the current targeted therapies (regorafenib, cabozantinib, and ramucirumab) target YAP, making YAP a promising novel target for drugs that may enhance treatment efficacy.

The YAP–Hippo signaling pathway has been reported to promote tumor development [46,47,48,49,50,51,52,53,54,55,56,57,58,59,60] as well as to confer drug resistance in a variety of cancers [19,20,23,25,52,61,62]. Overexpression of YAP is thought to be responsible for sorafenib resistance in some cancers including renal cancer [63] and liver cancer [25,26]. Several reports have shown that YAP is involved in the sorafenib resistance of HCC cells through hypoxia [26], cirrhotic stiffness [25], and upregulating surviving [64]. In the current study, we found significantly higher levels of YAP in both of the TCGA-liver cancer cell lines (Figure 1A) and in the tissues from HCC patients who did not respond to sorafenib (Figure 1B). Additionally, HCC cells that acquired sorafenib resistance showed significantly higher levels of YAP compared with naïve cells (Figure 1D). Blockage of YAP by YAP inhibitor synergistically increased the sorafenib sensitivity of the HCC-resistant cell lines (Figure 1E). These findings highlight a central role for YAP in the sorafenib resistance of HCC.

IGF-1R signaling plays an important role in a variety of human cancers. The IGF-1/IGF-1R/YAP pathway was reported to promote growth effects in triple-negative breast cancer (TNBC) cells [65]. However, the interplay between YAP and IGF signaling has remained uncertain in HCC sorafenib resistance. We previously illustrated that high expression of IGF-1R correlates with expression of cancer stemness markers (OCT4 and NANOG) and early recurrence [27,28] and with sorafenib resistance properties in HCC [11,66]. In the current study, we found that the underlying mechanism of HCC sorafenib resistance involves a YAP–IGF-1R signaling loop that involves EMT-related proteins and YAP nuclear translocation in vitro and in vivo. The results of our current in vitro (Figure 2A), in vivo (Figure 2B), TCGA-LIHC cohort (Figure 5A), and patient tissue (Figure 5B,C) studies all support a correlation of YAP with IGF-1R and EMT markers. In support, the suppression of YAP by a small molecular inhibitor or shRNA led to reduced expression of IGF-1R and EMT markers (Figure 3). Moreover, sorafenib-resistant HCC cells were induced to become sensitized to sorafenib in the presence of VP (Figure 1E). We also showed that the increased level of nuclear YAP is due to activation of IGF-1R (Figure 4B–E,H). Knockdown of IGF-1R by shRNA reduced the level of YAP in sorafenib-resistant cells (Figure 4F,G). Finally, using several small molecular inhibitors, we verified that the nuclear translocation of YAP was trigged through the IGF-1R/PI3K/mTOR transduction pathway (Figure 4I). Our findings highlight the significance of a YAP–IGF-1R signaling loop in HCC sorafenib resistance and provide novel potential targets for future therapeutics to treat patients with sorafenib-resistant HCC (Figure 6).

Stemness markers and EMT markers have shown that they are associated with nonspecific aggressive liver cancer. First, a number of reports have revealed the presence of stem-cell-like phenotype of cancer cells, referred to as tumor-initiating cells (TICs) [67]. The tumor microenvironment tightly responds to tumorigenesis [12], recurrence [27,68], metastasis [69], and chemoresistance [70] in the HCC patients. Meanwhile, the association of IGF-1R signaling in promoting cancer stemness properties has also been demonstrated in cancers. Besides, the role of IGF-1R signaling has been reported in tumorigenesis [71], drug resistance [72], and sorafenib resistance [12]. Second, EMT is considered to be a key process in driving tumor cell invasion and metastasis. The underlying mechanism of EMT, such as SNAIL family, TWIST family, and VIMENTIN, are all involved and play a key role in HCC progression [73]. Importantly, the stemness markers and EMT markers are related and complement each other to build the aggressiveness of HCC. It is often reported that cells with stemness properties could express high levels of EMT markers. In contrast, EMT also contributes to the increased population of cancer stem-like cells [74]. Therefore, the high expression of IGF-1R and EMT markers in sorafenib-resistant cells compared with naïve cells in the currently study has reinforced the evidence regarding the association of these genes with malignancy in liver cancer. High expression levels of these genes could predict poor diagnoses for patients with liver hepatocellular carcinoma (LIHC).

Biomarker screening has not been available to guide clinical trials of targeted drugs in HCC patients. Therefore, an unselected patient population may be the cause of the negative results of a ramucirumab phase III study [75]. Similarly, an unselected population was also enrolled in a clinical trial of linsitinib that also had negative results [76]. As HCC is well-known for its heterogeneity, selecting patients who are more likely to respond to specific targeted drugs may improve treatment outcomes. In the current study, we found high levels of YAP expression in the sorafenib-nonresponder BIOSTORM-cohort (Figure 1B) as well as in the nuclei of HCC tissue (Figure 5B) and in metastatic tissue (Figure 5C). These findings suggest that the tumor suppression effect of sorafenib is decreased in patients with overexpressed YAP. Hence, YAP may be a useful treatment target as well as a potential biomarker to use when selecting therapy for patients with HCC.

Combination therapy can improve treatment outcomes and is a growing trend in ongoing trials for advanced HCC [77]. YAP is a co-transcription factor that induces transcription of a variety of genes related to cell proliferation (E2F and Cyr61), cell survival (survivin and Bcl2), cancer stemness markers (Oct4, Nanog, and IGF-1R), and EMT markers [11,17,18,27,30,36,57,58,78,79,80,81,82,83,84]. Suppression of YAP can reduce the malignancy of many cancers [18]. Verteporfin, a YAP-specific inhibitor, is being studied in a phase I/IIa clinical trial for the treatment of primary breast cancer [85], a phase II trial for metastatic breast cancer [86], a phase I/II trial for glioblastoma [87], and a phase I trial for prostate cancer [88]. To the best of our knowledge, our study is the first one to demonstrate a role and mechanism of IGF-1R and YAP in sorafenib resistance. Our findings showing the synergist effect of verteporfin and sorafenib on sorafenib-resistant HCC cells suggest that the combination of YAP inhibitor and sorafenib may be of clinical use in HCC therapy.

## 5. Conclusions

A YAP–IGF-1R signaling loop appears to play a role in HCC sorafenib resistance. Treatment with IGF-1R ligand increases YAP expression; silencing IGF-1R effectively suppresses YAP protein levels in HCC cells. Blocking YAP using the specific inhibitor VP or YAP shRNA led to reduced expression levels of IGF-1R and EMT markers and re-sensitized sorafenib-resistant cells to sorafenib treatment. These findings demonstrate a role for a YAP–IGF-1R signaling loop in HCC sorafenib resistance. Targeting YAP–IGF-1R signaling may be an effective strategy for treating sorafenib-resistant HCC.

## Figures and Tables

**Figure 1 cancers-13-03812-f001:**
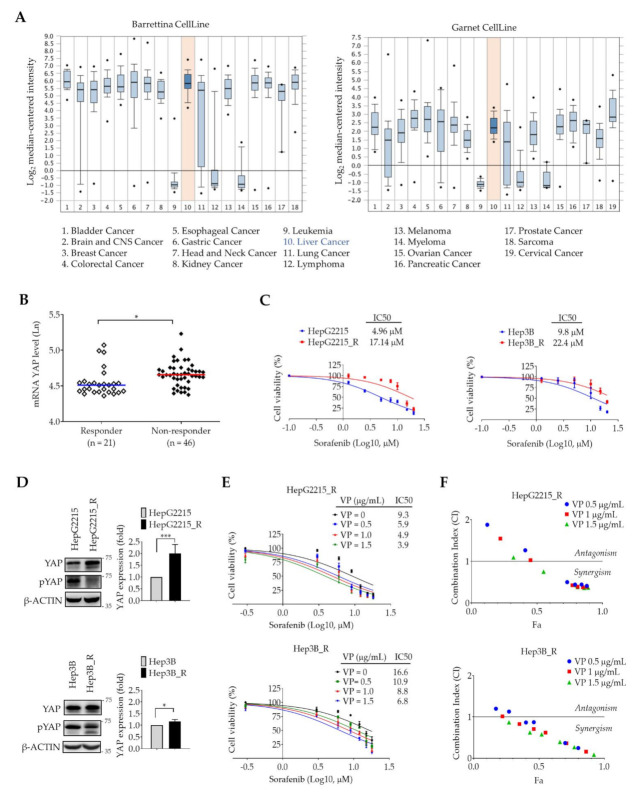
High expression of Yes-associated protein (YAP) correlates with sorafenib-resistant properties of hepatocellular carcinomas (HCCs). (**A**) mRNA expression levels of *YAP* in various cancer cell lines. Note the increase in *YAP* in liver cancer, from the Barretina CellLine database (3.6-fold, *p* = 1.22 × 10^−15^, left panel) and the Garnet CellLine database (2.1-fold, *p* = 1.27 × 10^−5^, right panel) through the ONCOMINE web server. Cancer types in the Barretina CellLine database are labeled 1–18, *n* = 917, Student’s *t*-test, and those in the Garnet CellLine database are labeled 1–19, *n* = 732, Student’s *t*-test. (**B**) mRNA levels of *YAP* in HCC tissues from patients treated with sorafenib (responders, *n* = 21; non-responders, *n* = 46). Mann–Whitney U test, * *p* < 0.05. (**C**) IC50 values of naïve HBV-HCC (HepG2215 and Hep3B) and sorafenib-resistant HBV-HCC (HepG2215_R and Hep3B_R) after sorafenib treatment (0, 1.25, 2.5, 5, 7.5, 10, 15, and 20 μM). (**D**) Protein levels of YAP and p-YAP in HepG2215 and HepG2215_R by Western blotting. Data are the mean ± SEM of at least three independent experiments. * *p* < 0.05 and *** *p* < 0.001 by Student’s *t*-test. (**E**,**F**) The cell viability assays of HepG2215 and Hep3B naïve/resistant cells treated with sorafenib (1, 5, 10, 15, 20 μM) with or without VP (0, 0.5, 1, 1.50 μg/mL) for 48 h through WST assay. Data are the mean ± SEM of at least three independent experiments. Calculated combination index (CI) values of (**E**). CI values were interpreted as follows: CI > 1, antagonistic effect; CI = 1, additive effect; and CI < 1, synergistic effect.

**Figure 2 cancers-13-03812-f002:**
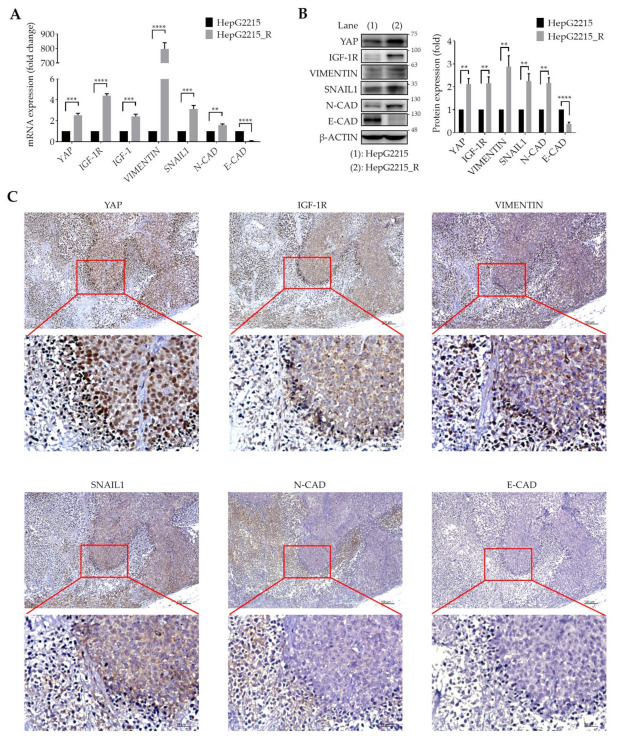
High association of YAP with IGF-1R and EMT-related proteins in sorafenib-resistant HCCs. Expression levels of (**A**) mRNA and (**B**) protein of YAP, IGF-1R, and EMT markers in naïve/resistant HepG2215 and HepG2215_R cells. Data are the mean ± SEM of at least three independent experiments. ** *p* < 0.01, *** *p* < 0.001, and **** *p* < 0.0001 by Student’s *t*-test. (**C**) Immuno-histochemical staining of expression and cellular localization of YAP, IGF-1R, and EMT markers (VIMENTIN, SNAIL1, and N-CAD) in xenograft of HepG2215_R tumor tissue in sorafenib-treated SCID mice. Scale bar, 100 µm.

**Figure 3 cancers-13-03812-f003:**
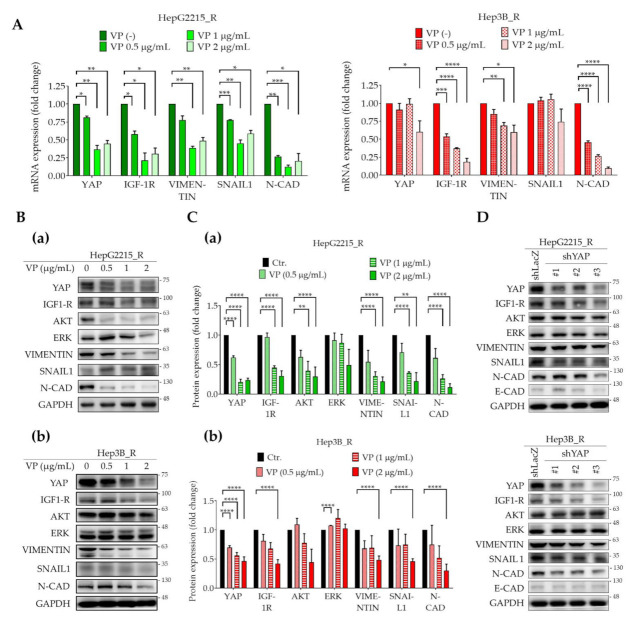
YAP regulates IGF-1R signaling-related proteins and EMT markers in sorafenib-resistant HCCs. (**A**–**C**) Effect of VP (a specific YAP inhibitor) on the expression levels of (**A**) mRNA and (**B**) protein of YAP, IGF-1R signaling-related proteins, and EMT markers in sorafenib-resistant HepG2215-R cells and Hep3B-R cells. (**C**) The quantitative data of (**B**). (**D**) Effect of YAP silencing by shRNA on the protein levels of YAP, IGF-1R-related signaling proteins, and EMT markers in HepG2215-R cells and Hep3B-R cells. For all quantification, data are the mean ± SEM of at least three independent experiments. * *p* < 0.05, ** *p* < 0.01, *** *p* < 0.001, and **** *p* < 0.0001 by Student’s *t*-test.

**Figure 4 cancers-13-03812-f004:**
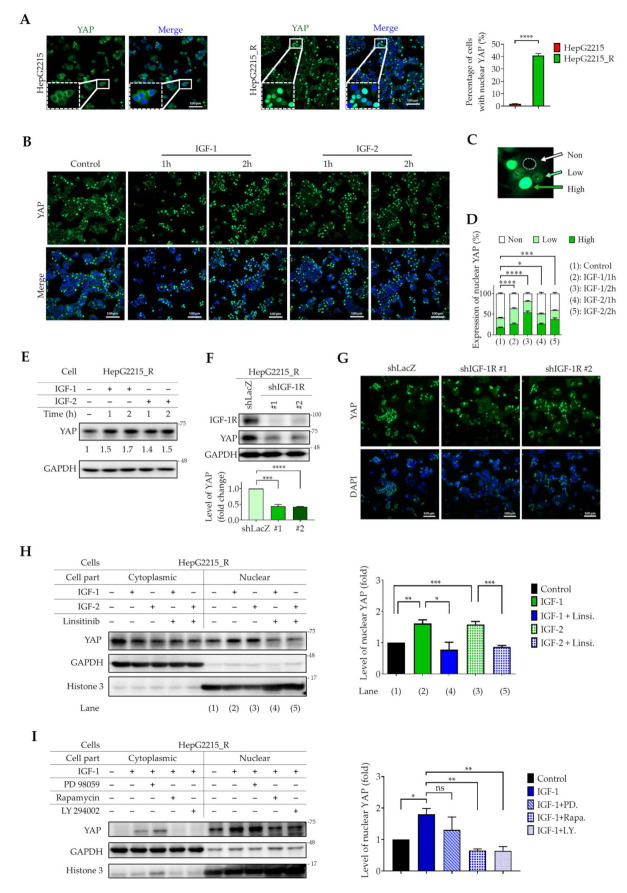
IGF-1R activation induces YAP nuclear translocation in sorafenib-resistant HCC cells. (**A**) Cellular localization of YAP in HepG2215 cells (left panel) and HepG2215_R cells (middle panel) by immunocytochemical staining. Cells with nuclear staining for YAP were quantified (right panel). Data are the mean ± SEM of at least nine random image fields. (**B**) The effect of IGF-1/2 (50 ng/mL, 1 h or 2 h treatment) on YAP nuclear translocalization in HepG2215_R cells. (**C**) The expression level of nuclear YAP in HepG2215_R cells was divided into three groups based on the fluorescence intensity: non, low, and high groups. (**D**) The quantitative analysis of (**B**). Data are the mean ± SEM of at least eight random image fields of **(B)** for each group. The effect of (**E**) IGF-1/2 (50 ng/mL, 1 h or 2 h treatment) or (**F**,**G**) IGF-1R silencing (shIGF-1R) on YAP protein expression in HepG2215_R cells by Western blotting assay and immunocytochemical staining. The relative quantification was normalized to the corresponding GAPDH. (**H**) The effect of IGF-1/2 (50 ng/mL, 1 h or 2 h treatment) with or without linsitinib (10 µM) on YAP expression in cytoplasmic and nuclear fractions of HepG2215_R cells. (**I**) The effect of signaling blockages targeting PI3K (ly294002), mTOR (rapamycin), and ERK (PD98059) on YAP nuclear translocation in HepG2215_R cells under IGF-1/2 treatment. The quantification of nuclear YAP from Western blotting is analyzed in the right panel. * *p* < 0.05, ** *p* < 0.01, *** *p* < 0.001, and **** *p* < 0.0001 by Student’s *t*-test.

**Figure 5 cancers-13-03812-f005:**
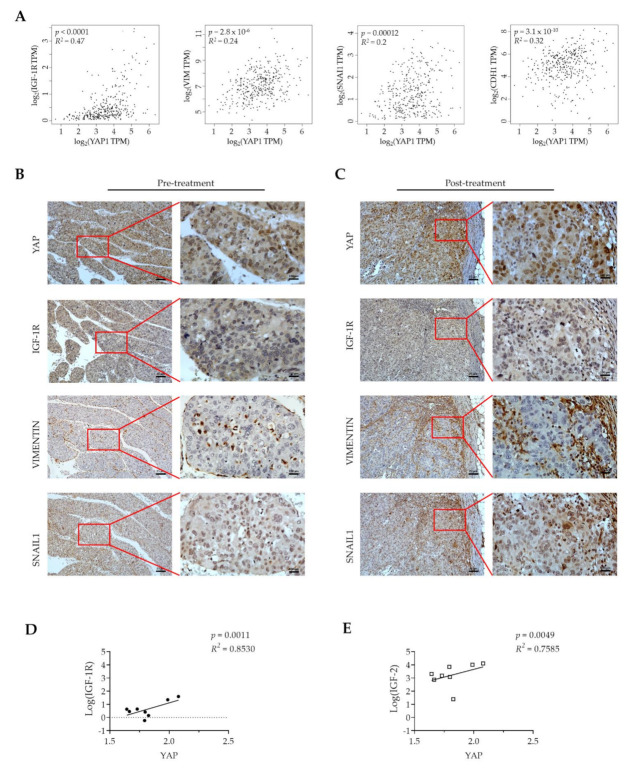
High expression of YAP is positively correlated with IGF-1R and EMT-related proteins in the tumor tissues of HCC patients who received sorafenib treatment. (**A**) Positive correlations of mRNA expression levels among *YAP*, *IGF-1R*, *VIMENTIN*, *SNAIL1*, and *E-CAD* (also known as CDH1) in the HCC cohort of TCGA database using the GEPIA web server. Pearson’s χ^2^ test. (**B**) The expression of YAP, IGF-1R, VIMENTIN, and SNAIL1 proteins in the HCC tissue from a patient pre-treatment and (**C**) 2 months post-sorafenib treatment through immunohistochemistry staining. The correlation of mRNA expression levels of *YAP* with (**D**) *IGF-1R* and (**E**) *IGF-2* in the TCGA-sorafenib treatment cohort (*n* = 8; Pearson’s χ^2^ test). *R*^2^, Pearson’s correlation coefficient.

**Figure 6 cancers-13-03812-f006:**
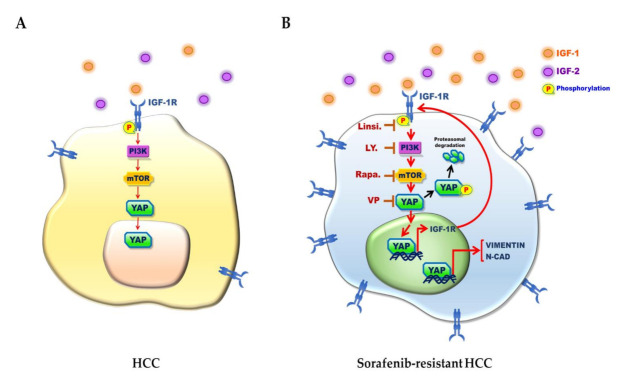
Schematic of the YAP/IGF-1R signaling loop involved in the sorafenib resistance and EMT property of HCC cells. (**A**) The naïve HCC cells show a lower expression level of YAP and IGF-1R. (**B**) The sorafenib-resistant HCC cells express higher levels of YAP and IGF-1R proteins. The IGF-1/2–IGF-1R signaling activation induces YAP nuclear translocation. YAP blockage by verteporfin (VP) reduces the expression levels of IGF-1R and EMT markers. The inhibition of IGF-1R signaling by small molecule inhibitors perturbs YAP nuclear translocation.

## Data Availability

The data presented in this study are available on reasonable request to the corresponding author.

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
