# Peer review of "A Yes-Associated Protein (YAP) and Insulin-Like Growth Factor 1 Receptor (IGF-1R) Signaling Loop Is Involved in Sorafenib Resistance in Hepatocellular Carcinoma"

_cancers, 2021, doi:10.3390/cancers13153812_

Round 1
Reviewer 1 Report
Specific comments:
- Neither the title nor the abstract fully reflects on the data sets.
- Lines 40-42: “Insulin-like growth factor-1 receptor (IGF-1R) is involved in the expression of epithelial-mesenchymal transition (EMT)-related proteins, which can induce drug resistance in cancers.” It is too vague to tighten in with the context.
- Lines 46-47: “synergistic combination index (CI)” was out of context.
- How did that the effects of sorafenib on IGF-1R signaling?
- How did they define naïve cells (HepG2215) and Hep3B)?
- Line 53: “LIHC patient tissues (46 nonresponders and 21 responders” – How did they define nonresponders or resistance? How did they differentiate them?
- HepG2215_R cells in SCID mice? Why did they not use immune-competent mouse models? How many mice per group? What’s the statistical power?
- Lines 54-55: “An in vivo model used sorafenib treatment of xenografts of HepG2215_R cells in SCID mice.” It is not a logical sentence.
- Lines 58-60: did they show it in vitro and in vivo?
- Lines 60-61: “The combination of YAP-specific inhibitor verteporfin (VP)
and sorafenib effectively decreased cell viability, with a synergistic combination index (CI)” – it is not logical.
- Lines 73 – 76, how many patients in either trial with demographics?
- Lines 77-83: How did YAP1 regulate “signaling pathways PI3K/AKT [6,7], Raf/Mek/ERK [8],Jak/Stat3 [9,10], cancer stemness [11,12], hypoxic environments, and EMT?”
- Lines 133-143: “Total mRNA was extracted from cells” – any patient’s tissues?
- Lines 149-159: any confocal microscopy for time-course studies to confirm in vivo side-by-side simultaneously?
- Lines 204-207: what’s the statistical power?
- Fig 1B: the error bars were overlapped. Why did they say it is statistically different? The same patterns were with cell lines with many outliers, which did not make liver cancer stand out.
- Fig1D: “Protein levels of YAP and p-YAP in HepG2215 and HepG2215_R b” – with WB data only, it is not accurate to state that protein level - YAP HepG2215_R was overexposed showing two bands.
- Fig2A vs. Fig 2B: YAP mRNA levels were not consistent with their counterpart protein levels in either cell line. Why is that?
- Fig 3. Any pYAP? Any drug-driven changes? Any pIGF-1R changes (Fig 6)? Fig 3B/C really can’t be quantified based on WB overexposure.
- Fig 4. The resolution of the images could not tell the translocation. Some co-localized biomarkers should be applied. Neither the quantification of nuclear YAP from western blotting is sufficient.
- Fig 5. Out of 67 patients, how many did they check? They did not justify Pearson’s χ2 test in Methods.
- Lines 383-386: “Based on previous data demonstrating” – did they show here again the old data?
- Any data did they have for patients applying verteporfin (or regorafenib, cabozantinib, and ramucirumab) and sorafenib? Why not?
- Fig 6. Schematic of YAP-IGF-1R signaling loop. Where did they show the loop? The current drawing of the scheme indicates only a linearly one-way relationship. Neither could the drawing capture their data set as presented. E.g., how did IGF-1R phosphorylation affect the pathway?
Reviewer 2 Report
In this paper, the authors describe a role of YAP in the sorafenib resistance in HCC. They demonstrate that inhibiting YAP could sensitize HCC to sorafenib. This is an interesting work giving insights into HCC resistance to sorafenib.
Comments:
-spell check is required, minor errors throughout the text
Fig 2C: It would be nice to be able to compare with naive cells to see if expression of these proteins also increases in vivo. Do the authors have performed that experiment as well?
Figure 3A: do the authors have an hypothesis to explain why VP seems not as effective in Hep3B cells?
Regarding sorafenib resistance, is it most likely an acquired resistance or innate resistance?
Reviewer 3 Report
This results of in vitro and xenograft assay of this paper are interesting. However, only samples from one single patient was used to show the interaction of YAP and IGF-1R in the EMT process after sorafenib treatment. The number of patients here is too low. The author may analyze more results from published database to make the conclusion more solid.
Reviewer 4 Report
- The study on the adjuvant setting of sorafenib (STORM) failed. In this study, after using sorafenib as an adjuvant, a case with recurrence was considered a non-responder, and a case without recurrence was considered a responder. It needs some background explanation.
- Since the expression levels of YAP, IGF-1R and mesenchymal markers are high in non-responders, can it be concluded that they are resistant to sorafenib? It is thought that the increase in their expression level may express the degree of aggressiveness of HCC. Can't it be considered that their over-expression may indicate nonspecific aggressiveness that is not related to sorafenib of HCC?
- As a minor point, in Result 3.2 (Figure 2C), it is the IHC result that the expression of five proteins tends to be expressed in the cancer cell margin. In the case of N-CAD, it seems to be evenly expressed in a region other than Cancer, can it be considered that it is expressed in margin?
Round 2
Reviewer 1 Report
accepted.
Reviewer 3 Report
My concern was addressed.
Reviewer 4 Report
It seems that the comments are well reflected.